# Use of Specific T Lymphocytes in Treating Cytomegalovirus Infection in Hematopoietic Cell Transplant Recipients: A Systematic Review

**DOI:** 10.3390/pharmaceutics16101321

**Published:** 2024-10-11

**Authors:** Tayná F. G. S. Bandeira, Luciana C. Marti, Edna T. Rother, Lucas Reis Correia, Clarisse M. Machado

**Affiliations:** 1PROADI-SUS, Hospital Israelita Albert Einstein, Sao Paulo 05652-900, SP, Brazil; lucas.rc@einstein.br; 2Instituto Israelita de Ensino e Pesquisa, Hospital Israelita Albert Einstein, Sao Paulo 05652-900, SP, Brazil; luciana.marti@einstein.br (L.C.M.); edna.rother@einstein.br (E.T.R.); 3Laboratório de Virologia, Instituto de Medicina Tropical, Faculdade de Medicina, Universidade de São Paulo, Sao Paulo 05403-000, SP, Brazil; clarimm@usp.br

**Keywords:** specific T lymphocytes, hematopoietic cell transplant, cytomegalovirus infection

## Abstract

Cytomegalovirus (CMV) poses a significant threat to post-hematopoietic cell transplantation (HCT). Control strategies include letermovir prophylaxis or ganciclovir pre-emptive therapy (PET). Without prophylaxis, 65–90% of seropositive recipients develop a clinically significant CMV infection. Due to PET drawbacks, letermovir prophylaxis is preferable, as it reduces CMV-related events and improves overall survival. However, refractory or resistant CMV-CS remains a challenge, with maribavir showing limited efficacy. This systematic review followed the Cochrane Manual and PRISMA guidelines and was registered in PROSPERO. Searches were conducted in PubMed, Scopus, Embase, and Web of Science. Out of 1895 identified records, 614 duplicates were removed, and subsequent screening excluded 1153 studies. Eleven included studies (2012–2024) involved 255 HCT recipients receiving adoptive immunotherapy (AI), primarily CMV-specific T-cell therapy. GvHD occurred in 1.82% of cases. Adverse events occurred in 4.4% of cases, while mild CRS was observed in 1.3% of patients. Efficacy, evaluated in 299 patients across eleven studies, showed an average response rate of 78.2%. CMV-CS recurrence was observed in 24.4% of 213 patients, and death due to CMV was reported in 9.7% of 307 patients across nine studies. Adoptive hCMV-specific T-cell immunotherapy appears to be a safe, effective alternative for refractory CMV-CS in HCT.

## 1. Introduction

Cytomegalovirus (CMV) is the most significant pathogen to occur after hematopoietic cell transplantation (HCT) [1]. Letermovir prophylaxis or ganciclovir (GCV) pre-emptive therapy (PET) are currently the main strategies to control CMV-related events after HCT. If no prophylaxis is used, 65% to 90% of the seropositive HCT recipients will develop a clinically significant CMV infection (CMV-CS) [2,3]. Currently, CMV-CS is typically defined as the presence of CMV viremia and/or disease requiring antiviral therapy [4].

In the last three decades, PET guided by antigenemia or quantitative PCR (qPCR) has been mostly used to control CMV [5,6], despite the inconveniences of GCV or foscarnet toxicities, CMV recurrences (≥30%), refractory or resistant CMV (11%), and the survival disadvantage observed in patients who develop CMV-CS [3,7].

Letermovir prophylaxis has been considered a more promising alternative to avoid CMV-CS and the main limitations of PET [8,9]. Besides reducing CMV-CS, letermovir prophylaxis decreases the frequency of CMV recurrences and the incidence of refractory or resistant CMV-CS, reduces non-relapse mortality (NRM), and increases overall survival (OS) [10,11,12].

Despite the improvements in CMV management after HCT, refractory/resistant CMV-CS with high mortality rates still occurs [11]. Maribavir, a recently approved antiviral for the treatment of refractory/resistant CMV-CS, showed limited efficacy (56%) and, therefore, around 40% of patients with refractory/resistant CMV will need additional therapy [13].

Currently, the only alternative left in cases non-responsive to Maribavir is adoptive T-cell immunotherapy. Virus-specific cell therapy in the context of HCT has a beneficial impact on both avoiding CMV reactivation as well as promoting the control of CMV-CS through the restoration of specific cellular immunity [14].

This article aims to report a systematic review of the literature on whether the use of adoptive hCMV-specific T-cell immunotherapy (AI) is safe and effective for the treatment of refractory CMV-CS in HCT recipients.

## 2. Materials and Methods

### 2.1. Strategy for the Search and Selection of Studies

This systematic review was conducted in accordance with the Cochrane Manual of Systematic Reviews, reported in accordance with the Preferred Reporting Items for Systematic Reviews and Meta-Analyses (PRISMA) [15], and registered in PROSPERO (International Prospective Register of Systematic Reviews) [16,17] under registration number CRD42023440361. We searched the PubMed, Scopus, Embase, and Web of Science databases on 19 June 2023. The search was updated on 17 July 2024, and two additional articles were included in the update of the search, meeting the criteria established in the PICOS. The detailed search strategy can be found in Appendix A. There were no restrictions on language or year of publication. The articles were imported into the Mendeley reference manager and duplicates were eliminated. The screening and selection were carried out on the Rayyan platform. Two reviewers (TFGS Bandeira and CM Machado) separately selected the title and abstract of each identified record, and disagreements were resolved by a third reviewer (LC Marti). Then, the full text was reviewed for all records that met the research eligibility criteria.

The inclusion criteria were as follows: (1) population: HCT patients with CMV-CS who were ineligible or refractory to conventional therapies; (2) intervention: adoptive hCMV-specific T-cell immunotherapy (AI); (3) restrictions: no restriction was presented to the comparator; (4) results: response rate, clearance or CMV viral load decrease (%), adverse events (AE) post-infusion, cytokine release syndrome (CRS), graft versus host disease (GvHD), CMV-CS recurrences after AI, and death due to CMV; (5) the design of the studies included in the analysis: randomized clinical trials (RCTs or NRCTs) and observational studies that presented a comparator or single-arm study that were original.

Exclusion criteria: (1) use of the intervention for prophylactic purposes; (2) patients who were not refractory or ineligible to conventional therapies; (3) abstracts of congresses or protocols of clinical trials or reviews; (4) narrative reviews were only considered for manually searching for potential articles.

### 2.2. Assessment of the Quality and Risk of Bias of the Included Studies

We used the Cochrane Risk of Bias in Non-Randomized Intervention Studies Tool (ROBINS-I) [18] to evaluate the included studies. This tool assesses the risk of bias in seven domains. The risk evaluation was classified as either low, moderate, severe, critical risk of bias, or no information.

The overall certainty of the evidence was assessed using the Grading of Recommendations Assessment, Development, and Evaluation (GRADE), and was classified as high, moderate, low, or very low evidence [19].

### 2.3. Data Extraction

According to the inclusion and exclusion criteria, data were extracted from all eligible studies and consolidated in the Microsoft Excel^®^ platform, including the first author, year, country of the study, title, design, comparator, intervention, population, sample size, outcomes, and funding sources. Clinical outcomes were as follows: efficacy (response rate and clearance or decrease in CMV viral load), safety (appearance or exacerbation of GVHD, AE after infusion, and CRS), and CMV recurrences after infusion of AI. Established definitions for clinical trials (CMV-CS, CMV recurrence, and DNAemia quantification) were used as references to analyze the efficacy of outcomes. The frequency of each clinical outcome was expressed in percentage (number of patients affected/total number of patients receiving CMV-specific AI).

## 3. Results

### 3.1. Included Studies

In total, 1498 records were identified in the initial search across the four databases, and 559 were removed as duplicates. The search was updated, identifying 397 records, and eliminating 55 duplicates. The subsequent title and abstract screening led to the exclusion of 1153 studies. Eleven studies were included in the analysis (Figure 1). The list of excluded studies is presented in Appendix A.

### 3.2. Characteristics of the Included Studies

The characteristics of the included studies are shown in Table 1. The funding sources for the studies included are presented in Appendix A. We included 11 studies [20,21,22,23,24,25,26,27,28,29,30] published between 2012 and 2024. In total, seven were NRCTs, four retrospective observational studies, and one a prospective observational study. The articles were from three countries: four from the USA, three from China, and one from Hungary. Four studies were multicenter. All studies were non-comparator and performed an analysis of the intervention only. Considering all studies included in this review, 373 HCT recipients received adoptive immunotherapy, 333 of whom received CMV-specific T cell therapy for refractory or resistant CMV infection or disease. Forty patients in two studies received EBV, ADV or ADV, and CMV-specific T cells [23,28,30].

### 3.3. Assessment of the Quality and Risk of Bias of the Included Studies

The analysis of the risk of bias in the incorporated studies is represented in Figure 2. One study showed critical global bias, six showed serious global bias, while four showed a moderate risk of global bias in the evaluation of their outcomes. The analysis of GRADE and its explanations are presented in Appendix A. The outcomes related to AE post-infusion, CRS, GvHD, and CMV-CS recurrence after AI showed low overall certainty of evidence.

The level of certainty of the evidence was moderate for the outcome response rate, clearance or VL reduction (%), and death related to CMV.

### 3.4. Outcomes

Table 2 provides a summary of the results found in the studies’ outcomes relating to effectiveness, safety, CMS-CS recurrence after AI, and death due to CMV, including the number of treated patients, cell origin, dosage, and number of doses. Three hundred thirty-three patients were included in the studies, ranging from seven to one hundred four. The origin of the cells for AI was from primary donors (*n* = 60), third-party donors (*n* = 263), or both (*n* = 3). Information about cell origin was not available in one study (*n* = 7). Dosages varied among the studies. Appendix A presents the number of doses, dosage, and cell sources for each included study.

#### 3.4.1. Safety

GvHD was evaluated in ten studies [20,22,23,24,25,26,27,28,29,30]. Among the 328 patients analyzed for post-infusion GvHD, seven (1.82%) developed the condition following AI. One patient, who initially had grade 1 GvHD, progressed to grade 2 [25]. Four patients developed grade 2 GvHD, one patient developed acute GvHD, and one patient experienced secondary rejection associated with VST expansion, which was thought to be related to the VST infusion.

Post-infusion adverse events: Nine of the eleven studies [22,23,24,25,26,27,28,29,30] evaluated adverse events after infusions in a total of 224 patients. Nine events (4.4%) possibly related to adoptive immunotherapy were reported.

CRS: Eight of the eleven studies [22,23,24,25,26,27,28] evaluated CRS in 157 patients. Only two cases of mild CRS (1.3%) were reported related to CMV-AI treatment [23,26]. In one study, a patient experienced mild dyspnea and hypoxia three days after infusion; HLA matching was not performed in this trial [23]. In another study, a patient developed a fever post-infusion [26]. The remaining studies reported no signs of CRS in their patients.

#### 3.4.2. Efficacy/Effectiveness

In this review, the efficacy/effectiveness of AI could be evaluated by the reported data on the response rate, and on the clearance or decrease in DNAemia. Eleven studies [20,21,22,23,24,25,26,27,28,29,30] evaluated this outcome in 299 patients. The average response rate was 78.2%.

CMV-CS recurrence after AI: Eight studies [20,21,22,23,25,26,27,28] evaluated this outcome. Out of 213 included patients, 52 patients (24.4%) experienced CMV-CS recurrence after adoptive immunotherapy, ranging from 0% to 42.8%.

Death due to CMV: Eight studies evaluated death due to CMV [21,22,23,25,27,28,29,30]. Two studies [25,28] did not report deaths associated with CMV. In six other studies [21,22,23,27,29,30], the total number of reported deaths due to CMV was 30 out of 307 patients (9.7%). One patient died of CMV encephalitis [22], four patients died from CMV-associated pneumonitis (three proven, one probable) [22,23,29,30], twenty-one patients died from CMV without further specifications [21,30], and three patients were non-responders and died from disease progression [27].

## 4. Discussion

With the arrival of letermovir, a safe and effective oral antiviral for CMV prophylaxis [10], a great improvement in the management of CMV-CS has been observed in the setting of HCT. However, if prophylaxis fails, the patient may develop repeated episodes of CMV-CS, which may favor the emergence of refractory CMV-CS and antiviral resistance.

Although letermovir has temporarily occupied the space of adoptive therapy in the prophylaxis scenario, the same did not occur in the treatment scenario of refractory or resistant CMV-CS, as even the best current alternatives may present limitations in toxicity and efficacy.

We conducted this systematic review, which included nine studies, to evaluate the safety and efficacy of adoptive immunotherapy for the treatment of HCT recipients with refractory CMV-CS. These studies were single-arm observational and NRCT phase I or II, resulting in penalties in domains associated with confounding bias.

Since the pioneering studies demonstrating the possibility of reconstitution of cellular CMV immunity in recipients of allogeneic HCT by the transfer of T-cell clones from donors [31], several studies have emerged evaluating the efficacy and safety of adoptive immunotherapy to control CMV-CS [32,33,34,35,36,37].

In general, adoptive immunotherapy for CMV prophylaxis or the treatment of refractory CMV-CS is safe and effective. However, access and cost are the biggest limitations to its expanded use.

The present systematic review has several limitations that will be addressed below. Among the included studies, six [20,22,23,24,26,29] presented a serious risk of bias in the results evaluated, four studies [21,27,28,30] presented a moderate risk of bias in the results, and one study [25] presented a critical risk of bias.

One of the main challenges of this systematic review was the heterogeneity of the data generated by the few selected studies. Most of them were NRCT phase I or phase II studies, retrospective or prospective observational studies that did not use any other therapy as a comparator group to AI, and lacked randomization, resulting in penalization in the domains associated with confounding bias.

Due to access limitations, the preparation and use of AI have been primarily restricted to research centers. Consequently, there is still no consensus on various aspects, including the source of clones (donor or third parties), the concentration of cells to be infused, criteria for defining response and failure, the persistence of transferred cells, the required number of infusions, and definitions of adverse events, among other factors.

According to GRADE, we observed low overall certainty of the evidence for the outcomes of GvHD, post-infusion AEs, CMV-CS recurrence after AI, and CRS. The level of certainty of evidence was moderate for the outcomes of death due to CMV and effectiveness (response rate, clearance, or reduction in VL). Indeed, data strength and objectivity are lost in the absence of reference standards. For instance, how to evaluate safety without strict and consensual definitions of adverse events? Or clinical response, without standardized dosage, number of infusions, or response criteria?

Acute or chronic GvHD is a common complication of allogeneic HCT. When evaluating the safety of AI, it is crucial to distinguish between pre-existing GvHD and GvHD that arises as a result of AI infusion. In this review, one study did not specify whether GvHD occurred as a consequence of AI [21]. In the remaining ten studies, the incidence of new or exacerbated GvHD following AI infusion was 1.82%.

In more detail, Koehne et al. [22] reported one patient with grade 3 GvHD who was undergoing treatment at the time of AI, with no flare-ups or new cases of GvHD following the infusion. Another study [30] found that 21% of patients had a history of GvHD, but no de novo GvHD was observed after adoptive immunotherapy (AI), though four patients experienced flares of pre-existing acute GvHD (aGvHD). Neuenhahn et al. [24] described two cases of GvHD: one involving a flare-up of pre-existing GvHD, and the other a new onset of grade 2–3 GvHD. The latter patient had previously received an NK cell transfusion and an unselected DLI before AI. Another study [25] reported one case of pre-existing grade 1 GvHD that progressed to grade 2 acute GvHD after AI. Similarly, another study [26] documented three cases of pre-existing GvHD, with two patients having grade 2 and one having grade 3 GvHD, but no new cases occurred post-AI.

Withers et al. [28] reported two cases of acute GvHD, one grade 2 and one grade 4, in a patient who received AI for CMV. Prockop et al. [29] noted that, despite 26 patients having a history of GvHD, no flares occurred after AI infusion. Other authors did not report pre-existing or de novo GvHD cases.

Ruan et al. [27] found that patients without CMV infection had a lower incidence of grade 1–2 GvHD (16.9%) compared to those with CMV infection (26%). However, the opposite trend was observed for grade 3–4 GvHD, with more cases in patients without infection (10.6%) compared to those with infections (8.7%). Only grade 1–2 GvHD was associated with a higher incidence of CMV infection, but no GvHD progression was observed in patients receiving AI.

Preemptive donor lymphocyte infusion (DLI) administered for minimal residual disease (MRD) after allogeneic HCT in acute leukemia has been associated with an overall incidence of acute GvHD at 12% and chronic GvHD at 31% [38]. In the absence of a direct comparator, the 3% incidence of GvHD as an adverse event of AI could be considered relatively low when compared to GvHD rates following DLI. This difference is likely because DLI contains alloreactive T cells, whereas CMV-specific cytotoxic T lymphocytes (CMV CTLs) are expected to target only CMV-infected cells.

DNAemia is a solid biomarker of CMV-CS; however, its use as an indicator of treatment failure or CMV-CS recurrence must comply with strict definitions and quantification criteria [39], which vary between HCT centers.

In this review, most of the studies used quantitative PCR (qPCR) to monitor AI response. However, the ideal conditions for accurate results of CMV qPCR were hardly met. Although not specifically stressed in the text, apparently only two studies [21,26] used commercial CMV qPCR assays and expressed viral load results in international units per milliliter (IU/mL). Few studies mentioned the type of sample (plasma, whole blood, or other fluid) taken for qPCR, and most of the studies used different DNAemia cut-offs or no cut-off at all [20,22,23,24,25,26,27,28,29,30], or used two different techniques, pp65 antigenemia and qPCR, to define AI response or CMV recurrences [20,22].

As previously demonstrated, CMV DNAemia quantification varies according to the qPCR protocol (in-house or commercial), the type of sample (plasma or whole blood), and the DNA extraction platform, not to mention intra- and interlaboratory variation. The use of the CMV international standard for CMV DNA quantification developed by the National Institute for Biological Standards and Control (NIBSC), with the results expressed in IU/mL, helps to diminish the variability among tests [40].

According to the present review, the overall response rate to AI in treating refractory or resistant CMV-CS was 76%, ranging from 57.1% to 90%. Currently, the best option for treating refractory or resistant CMV is maribavir, with a response rate of around 57% [41]. Therefore, although still limited by cost and access, adoptive immunotherapy seems to be a more effective alternative.

Concerning the outcome of “CMV recurrence”, this review showed a moderate level of certainty of evidence. The heterogeneity of definitions regarding CMV recurrences may explain this finding. CMV recurrence is defined as new CMV-CS in a patient with previous evidence of CMV-CS in whom the virus has not been detected for at least 4 weeks during CMV monitoring [39]. Only one article defined recurrence according to this accepted definition [27]. In the remaining ten studies, CMV recurrence was not properly defined [20,21,22,23,25,26,28] or was not reported [24,29,30].

No matter the approach used to control CMV-CS, CMV recurrences are considered failures of the CMV-CS control strategy, generally due to the lack or instability of CMV-specific immune reconstitution. The 24% rate of CMV recurrence after AI observed in this review is similar to the rates observed with other strategies. Recent studies have shown rates of CMV recurrences ranging from 20% to 38% in patients who received GCV preemptive therapy [3,42].

Death due to CMV is the undesirable consequence of treatment failure in refractory or resistant CMV-CS. A previous study, evaluating the burden of refractory CMV-CS in transplant populations treated with ganciclovir, foscarnet, or cidofovir, showed lethality rates of over 30% [43]. In the HCT setting, lethality rates of 1.7% were observed in patients with refractory/resistant CMV receiving maribavir [43]. In the present review, death due to CMV occurred in around 12% of the patients, highlighting the importance and severity of refractory/resistant CMV-CS in allogeneic HCT recipients.

With the recent progress of cellular therapies, leading to more elaborate and specific AI products [44], and the development of commercial platforms to increase access to AI, there is an urgent need that the proposed CMV definitions for use in clinical trials be universally adopted.

## 5. Conclusions

In conclusion, AI has proven to be a safe and attractive alternative for the treatment of refractory/resistant CMV-CS in HCT, showing better response rates compared to maribavir, with similar rates of CMV recurrence.

## Figures and Tables

**Figure 1 pharmaceutics-16-01321-f001:**
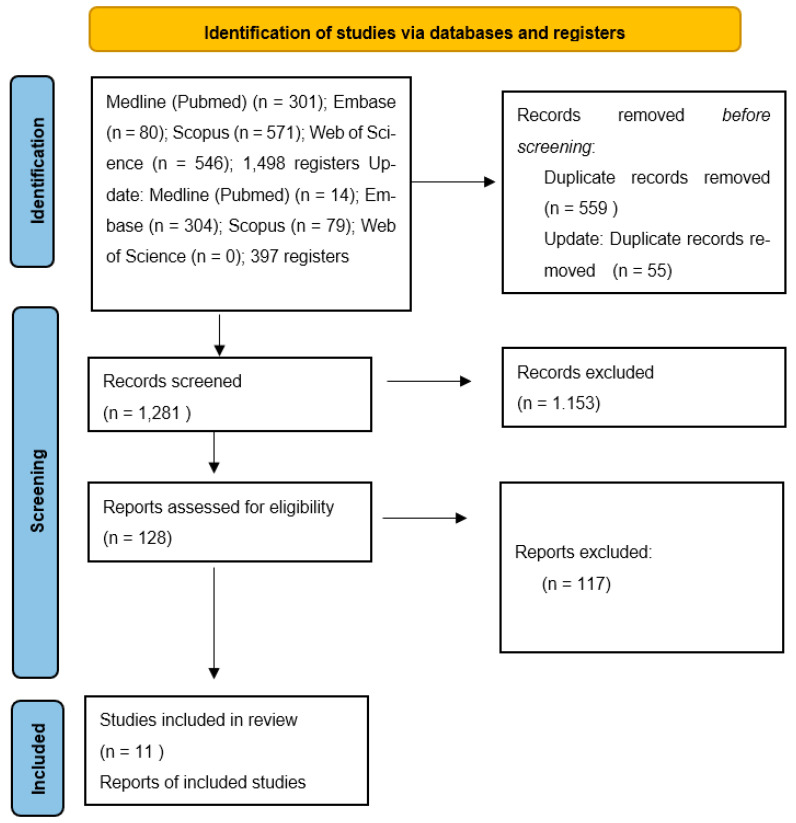
Study selection flowchart.

**Figure 2 pharmaceutics-16-01321-f002:**
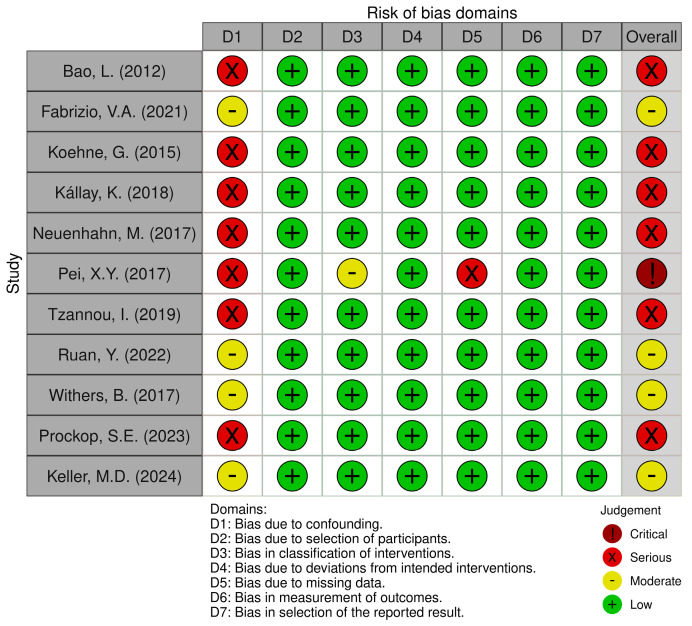
Risk of bias of the included articles according to ROBINS-I [20,21,22,23,24,25,26,27,28,29,30].

**Table 1 pharmaceutics-16-01321-t001:** Characteristics of the included studies.

First Author, Year	Country	Study Design	Intervention	Population (n)	Outcomes
Bao, L. (2012) [20]	Multicenter	NRCT phase I	hCMV-specific T cells	Allogeneic HCT patients with resistant or refractory CMV-CS (*n* = 7)	GvHD, DNAemia, death from other causes
Fabrizio, V. A. (2021) [21]	USA	Retrospective observational study	hCMV-specific T cells	Allogeneic HCT recipients with persistent CMV infection or refractory to antiviral therapy (*n* = 104)	GvHD, death from other causes, death due to CMV, OS, response rate
Koehne, G. (2015) [22]	USA	NRCT phase I	hCMV-specific T cells	Allogeneic HCT recipients with CMV DNAemia or persistent infection despite drug use (*n* = 16)	GvHD, AE post-infusion, DNAemia, death from other causes, death due to CMV, CMV recurrence
Kállay, K (2018) [23]	Hungary	Retrospective observational study	CMV or EBV or AdV-specific T cells	Pediatric allogeneic HCT patients with progressive or uncontrolled viral disease despite antiviral treatment (*n* = 9)	GvHD, AE post-infusion, CRS, DNAemia, death from other causes, OS, death due to CMV
Neuenhahn, M (2017) [24]	Multicenter	NRCT	Streptacell-specific hCMV T cells isolated from streptamers	Allogeneic HCT recipient’s refractory to antiviral therapy (*n* = 17)	GvHD, AE post-infusion, DNAemia, death from other causes, death due to CMV, response rate
Pei, Xu-Ying (2017) [25]	China	Prospective observational study	hCMV-specific T cells	Haploidentical HCT recipients with refractory CMV-CS (*n* = 64)	GvHD, AE post-infusion, DNAemia, death from other causes, death due to CMV
Tzannou, I. (2019) [26]	USA	NRCT	hCMV-specific T cells	Allogeneic HCT recipients with persistent CMV infection or disease or refractory to antiviral therapy (*n* = 10)	GvHD, CRS, DNAemia, death from other causes, response rate
Ruan, Y. (2022) [27]	China	Retrospective observational study	hCMV-specific T cells	Allogeneic HCT recipients with recurrent or treatment-refractory infection and CMV disease (*n* = 29)	GvHD, AE post-infusion, death due to CMV, response rate
Withers, B (2017) [28]	Multicenter	NRCT	CMV or EBV or AdV-specific T cells	Allogeneic HCT recipients with persistent or recurrent CMV-CS after treatment (*n* = 30)	GvHD, AE post-infusion, DNAemia, death from other causes, death due to CMV, CMV recurrence, OS, response rate
Prockop, S. E. (2023) [29]	USA	NRCT	hCMV-specific T cells	Allogeneic HCT recipients with recurrent or treatment-refractory infection and CMV disease (*n* = 67)	Toxicities, adverse events, response rate, GvHD, death due to CMV
Keller, M. D. (2024) [30]	Multicenter	NRCT	Partially HLA-matched VSTs targeting cytomegalovirus, Epstein–Barr virus, or adenovirus	Pediatric patients with inborn errors of immunity and/or post-allogeneic HCT with refractory viral infections (*n* = 16 *)	Safety and clinical responses

Legend: GvHD: Graft-versus-host disease; hCMV: Human cytomegalovirus; CMV-CS: Clinically significant CMV infection; HCT: Hematopoietic cell transplantation; NRCT: Non-randomized clinical trial; USA: United States of America; EBV: Epstein–Barr virus; AdV: Adenovirus; CRS: Cytokine release syndrome; AEs: Adverse events; OS: Overall survival; VSTs: virus-specific T cells. (*) In the total number (51) of patients in Arm A (HCT), 24 patients exhibited Adv, 16 had CMV, 7 patients had Adv and CNV, and 4 had EBV.

**Table 2 pharmaceutics-16-01321-t002:** Results of the included studies.

Author, Year [Ref]	No of pts Receiving AI	Response Rate, Clearance or VL Reduction (%)	AE Post-Infusion	CRS	Number with GvHD (%)		CMV-CS Recurrence after AI	**Death Related to CMV (%)**
					At AI Infusion	After Infusion		
Bao, L. (2012) [20]	7	4 /7 CR (57.1) 2/7 PR (28.6) 1/7 NR (14.3)	Not mentioned	Not mentioned	1/7 cGvHD (14.3)	0 (0)	3/7 (42.8)	Not mentioned
Fabrizio, V. A. (2021) [21]	104	60/85 * CR/PR (70.6) 25/85 * NR (29.4) 19 not evaluable	Not mentioned	Not mentioned	1/104 cGvHD (0.96)	0 (0)	25/85 * (29.4)	20/104 (19.2) died from CMV
Koehne, G. (2015) [22]	17 (1 not evaluable)	16 received AI **; 14 evaluable for response 10/14 CR (71.4) 2/14 PR (14.3) 2/14 NR (14.3)	0 (0)	0 (0)	1/16 ** (6.25) aGvHD	0 (0)	2/16 ** (12.5)	2/16 ** (12.5) One died of probable CMV pneumonia, and one died of CMV encephalitis
Kállay, K. (2018) [23]	6	5/6 CR (83.3) 1/6 NR (16.6)	0 (0)	1/6 (16.6)	4/6 aGvHD (66.6)	0 (0)	0 (0)	1/6 (16.6) died of CMV pneumonitis
Neuenhahn, M. (2017) [24]	17	9/16 ** CR (56.25) 3/16 ** PR (18.75) 4/16 ** NR (25)	1/16 ** (6.25)	0 (0)	0 (0)	2/16 ** (12.5) 1 aGvHD 1 cGvHD	Not mentioned	1/16 ** (6.25) died possibly due to CMV pneumonia
Pei, X.Y (2017) [25]	32	27/32 CR (84.3) 1/32 PR (3.1) 4/32 NR (12.5)	0 (0)	0 (0)	1/32 (3.1) aGvHD	1/32 (3.1) exacerbation aGvHD grade 1–2	5/32 (15.6)	0/32 (0)
Tzannou, I. (2019) [26]	10	7/10 CR (70) 3/10 PR (30)	0 (0)	1/10 (10) One pt with fever 8 h after infusion	3/10 (30) aGvHD	0 (0)	1/10 (10)	Not mentioned
Ruan, Y. (2022) [27]	29	26/29 CR (89.6) 3/29 NR (10.3)	0 (0)	0 (0)	Not mentioned	0 (0)	11/29 (37.9)	3/29 (10.3) died from CMV
Withers, B. (2017) [28]	28	22/28 CR (78.6) 5/28 PR (17.8) 1/28 NR (3.6)	0 (0)	0 (0)	0 (0) aGvHD	2/28 (7.1) aGvHD	5/28 (17.8)	0/28 (0)
Prockop, S.E. (2023) [29]	67	20/59 # CR (33.9) 18/59 # PR (30.5) 21/59 # NR (35.6)	9 (13.4) ***	NR	0/67 (0)	1/67 (1.5) de novo aGvHD	NR	1/59 # (1.7) died of CMV pneumonitis
Keller, M.D. (2024) [30]	20	7/13 ## CR (53.8) 3/13 ## PR (23) 3/13 ## NR (23)	0 (0)	0 (0)	Not mentioned	1/13 ## (7.7) ****	NR	2/13 ## (15.4) 1 CMV-IP; 1 CMV disease
Total ###	337	234/299 CR/PR (78.2) 65/299 NR (21.7)	10/224 (4.4)	2/157 (1.3)	11/286 (3.8)	6/328 (1.82)	52/213 (24.4)	30/307 (9.7)

Legend: Pts: patient; AI: Adoptive immunotherapy; AEs: Adverse events; CRS: Cytokine release syndrome; GvHD: Graft-versus-host disease; aGvHD: acute GvHD; cGvHD: chronic GvHD; CR: complete response; PR: partial response; NR: No response; * 19 pts were not evaluable; ** One pt was not evaluable; *** Nine subjects had 21 possibly related adverse events; **** Secondary rejection was associated with VST expansion and felt to be related to VST infusion; # 8 pts were not evaluable; ## 7 pts were not evaluable; ### Total number of evaluable pts in each variable.

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
