# Peer review of "Use of Specific T Lymphocytes in Treating Cytomegalovirus Infection in Hematopoietic Cell Transplant Recipients: A Systematic Review"

_pharmaceutics, 2024, doi:10.3390/pharmaceutics16101321_

Round 1

Reviewer 1 Report

Comments and Suggestions for Authors

This is a nicely written systematic review article and authors did a great job of analyzing it. The inclusion of consort diagram and risk and bias chart is extremely useful for easy read. Following are some concerns:

1. Out of 128 article selected for reading only 11 studies were included out of which 4 were retrospective studies and only 1 prospective studies along with 4 non randomized clinical trial. Authors should mention about the possiblity of biases in these studies.

2. If the data is available it would be ideal to include the demographical information of these 11 studies along with any pre-existing conditions, disorders or co morbidities which can provide further resolution of AI benefits. 

Comments on the Quality of English Language

Fairly well written and articulated

Author Response

Comments 1: "Out of 128 article selected for reading only 11 studies were included out of which 4 were retrospective studies and only 1 prospective studies along with 4 non randomized clinical trial. Authors should mention about the possiblity of biases in these studies". Response 1: We agree with these comments. "We included 11 studies [19–29] published between 2 012 and 2 024. In total, 7 were NRCT, 4 retrospective observational studies, and 1 prospective observational study" [L118-L120, page 3, paragraph 3: Characteristics of the included studies]. Possible biases in the studies have been reported on page 5, paragraph 1, lines 136-138. They can also be found in Figure 2 on page 6. "The analysis of the risk of bias in the incorporated studies is represented in Figure 2. "One study showed critical global bias, six showed serious global bias, while four showed moderate risk of global bias in the evaluation of their outcomes."Furthermore, potential biases related to the studies were discussed in lines 224–227, page 8, paragraph 10 "The present systematic review has several limitations that will be addressed below. Among the included studies, six [19,21–23, 25,28] presented a serious risk of bias in the results evaluated, four studies [20,26,27,29] presented a moderate risk of bias in the results, and one study [24] presented a critical risk of bias". Thanks for pointing this out.

Comments 2: If the data is available it would be ideal to include the demographical information of these 11 studies along with any pre-existing conditions, disorders or co morbidities which can provide further resolution of AI benefits. Response 1:Agree. Thanks for pointing this out. We agree with these comments. In our manuscript we have the demographic data of the eleven studies as number of patients included (Table 2. Results of the included studies.), page 6 of the manuscript. Furthermore, in addition to this data, we have information about the cell of origin of the adoptive immunotherapy, the dosage, and the number of doses (Table S5. Summary of Doses, Dosage, and Cell Sources for Included Studies.), page 11. Therefore, we have added a column in Table S5, page 11 in the supplemental material with the initial diagnosis data for patients in the included studies.

Reviewer 2 Report

Comments and Suggestions for Authors

This is a welcome report summarizing studies using CMV specific T cells as therapy for CMV CS that is refractory to pharmaceutical therapy.This systemic review is performed in accordance with Cochrane requirements.

CM-CS should be better defined in the introduction even if this may vary from variouse reports.

L 46 Regarding high mortality,give a reference.

In Table 1 include overall survival in the variouse studies as an outcome.If available from the original reports give more details regarding acute GVHD.What was the proportion of the variouse grades of acute GVHD before and after CMV CTL therapy?There is a correlation between acute GVHD and CMV reactivation following HCT.

Regarding efficacy.Is it possible to give more details regarding possible clearance of CMV DNA from the variouse studies?CR?PR?NR?before and after CMV-CTL.

In the discussion comparing DLI with CMV CTL it should be noted that DLI contains alloreactive T cells whereas CMV CTL are supposed to be CMV specific.This is a major difference.

Comments on the Quality of English Language

Language is ok according to my reading,not being native english speaker.

kind regards

Olle

Olle Ringden MD PhD

Professor in Transplantation Immunology 

karolinska Institutet

Author Response

Comments 1- "CM-CS should be better defined in the introduction even if this may vary from various reports."

Answer:

Thank you for the suggestion. A definition of CMV-CS has been added on line 37.

Comments 2-"L 46 Regarding high mortality, give a reference."
Answer:
Please see this information is stated in the following reference, that was included in the article.
Ljungman P, Schmitt M, Marty FM, Maertens J, Chemaly RF, Kartsonis NA, et al. A mortality analysis of letermovir prophylaxis for Cytomegalovirus (CMV) in CMV-seropositive recipients of allogeneic hematopoietic cell transplantation. Clin Infect Dis. 2020;70(8):1525–33.

Comments 3- "In Table 1 include overall survival in the various studies as an outcome. If available from the original reports give more details regarding acute GVHD. What was the proportion of the various grades of acute GVHD before and after CMV CTL therapy? There is a correlation between acute GVHD and CMV reactivation following HCT."

Answer:

The GvHD occurrence was reported in the safety item page 169 and in Table 2. More detailed information was added to Table 2, and the following text was included in the discussion section Line 254.
“In more detail, Koehne et al. [22] reported one patient with grade 3 GvHD who was undergoing treatment at the time of AI, with no flare-ups or new cases of GvHD following the infusion. Another study [30] found that 21% of patients had a history of GvHD, but no de novo GvHD was observed after adoptive immunotherapy (AI), though four patients experienced flares of pre-existing acute GvHD (aGvHD). Neuenhahn et al. [24] described two cases of GvHD: one involving a flare-up of pre-existing GvHD, and the other a new onset of grade 2-3 GvHD. The latter patient had
previously received an NK cell transfusion and an unselected DLI before AI. Another study [25] reported one case of pre-existing grade 1 GvHD that progressed to grade 2 acute GvHD after AI. Similarly, another study [26] documented three cases of pre-existing GvHD, with two patients having grade 2 and one having grade 3 GvHD, but no new cases occurred post-AI.
Withers et al. [28] reported two cases of acute GvHD, one grade 2 and one grade 4, in a patient who received AI for CMV. Prockop et al. [29] noted that, despite 26 patients having a history of GvHD, no flares occurred after AI infusion. Other authors did not report pre-existing or de novo GvHD cases. Ruan et al. [27] found that patients without CMV infection had a lower incidence."
Ruan et al. [27] found that patients without CMV infection had a lower incidence of grade 1-2 GvHD (16.9%) compared to those with CMV infection (26%). However, the opposite trend was observed for grade 3-4 GvHD, with more cases in patients without infection (10.6%) compared to those with infections (8.7%). Only grade 1-2 GvHD was associated with a higher incidence of CMV infection, but no GvHD progression was observed in patients receiving AI. “

Comments 4- "Regarding efficacy. Is it possible to give more details regarding possible clearance of CMV DNA from the various studies? CR? PR? NR? before and after CMV-CTL."

Answer:

When available, the detailed information about CR, PR and NR response to CMV-CTL infusion was included in Table 2

Comments 5- "In the discussion comparing DLI with CMV CTL it should be noted that DLI contains alloreactive T cells whereas CMV CTL are supposed to be CMV specific. This is a major difference."

Answer:

This sentence was re-written to include this information.

Reviewer 3 Report

Comments and Suggestions for Authors

In this manuscript, Authors present a systematic review of the literature on the use of adoptive hCMV-specific T-cell immunotherapy (AI) for the treatment  of refractory CMV-CS in HCT recipients. They summary, that AI is a safe and attractive alternative for the treatment of refractory/resistant CMV-CS in HCT. The topic is important for clinicians, and a paper is interesting and well written. Tables and figures are clear. References are current.

In my opinion, this manuscript is suitable for Readers of this journal.

Author Response

Agree. We greatly appreciate the positive feedback and recognition of the importance of our work on adoptive hCMV-specific T-cell immunotherapy for the treatment of refractory CMV-CS in HCT recipients. We are pleased to hear that the manuscript is considered interesting and well-written, with clear tables and figures, and current references.

No specific changes were requested by the reviewer. However, we have thoroughly reviewed the manuscript to ensure clarity and precision in the presentation of our findings. We have also verified that all tables and figures are accurately referenced and clearly presented.

Thank you once again for your encouraging comments and for recognizing the relevance of our study for the readers of Pharmaceutics Journal.

[No updated text in the manuscript was necessary based on this reviewer's comments.]
